# Bayesian Meta-Analysis of Myopia Control with Multifocal Lenses

**DOI:** 10.3390/jcm10040730

**Published:** 2021-02-12

**Authors:** Saulius Varnas, Xiaomeng Gu, Andrew Metcalfe

**Affiliations:** 1Carl Zeiss Vision Australia Holdings Ltd., 1284 South Rd., MAB G13/6 MAB Eastern Promenade, Tonsley, SA 5042, Australia; 2School of Mathematical Sciences, University of Adelaide, North Terrace, Adelaide, SA 5005, Australia; xiaomeng.gu@adelaide.edu.au (X.G.); andrew.metcalfe@adelaide.edu.au (A.M.)

**Keywords:** juvenile myopia, intervention studies, multifocal spectacles, multifocal contact lenses, meta-analysis

## Abstract

The aim of this study is to provide reliable guidelines for the mean percentage efficacy together with the 95% credibility interval in slowing down progression of myopia by a specific intervention over defined time periods, derived from a substantial number of randomised controlled clinical trials (RCTs) with consistent outcomes. Multifocal spectacles and contact lenses represent interventions with the largest number of RCTs carried out. Our meta-analyses considered 10 RCTs involving 1662 children which have tested the efficacy of progressive addition spectacle lenses (PALs). In a separate model for comparison purposes nine RCTs with 982 children trialling soft multifocal contact lenses (MFCLs) were analysed. Bayesian random-effects hierarchical models were fitted. The highest efficacy in retarding progression of the scaled sphere equivalent refraction was achieved after 12 M follow-up with the mean 28% reduction in progression and the 95% credibility interval between 21% and 35%. For comparison, the 95% credibility interval for the mean efficacy of soft MFCLs at 12 M follow up is 21% to 37%. We conclude that both multifocal spectacle and contact lenses moderately slow down progression of myopia, relative to single-vision spectacle lenses (SVLs) in the first 12 months after intervention. The relative efficacy of PALs tends to weaken after the first 12 months.

## 1. Introduction

High levels of myopia are a serious threat to public health due to the pathological changes in the retina and choroid associated with the excessive axial elongation of the eyeball [1,2,3,4]. There is an increasing prevalence of juvenile myopia, especially in East and Southeast Asia, [5,6,7,8] but also in Europe [9] and the USA [10]. High myopia is associated with the development of macular degeneration, retinal detachment, glaucoma, myopic retinopathy and premature cataracts [1,11].

There may be several causes for the progression of myopia. The most prevalent form of myopia, that is the main cause of the recent rises in prevalence of myopia worldwide, is called school myopia, and affects school children and students. It has been suggested that this form of myopia is primarily affected by environmental factors, such as prolonged near work and not enough time spent outdoors, and occurs as a disease of advanced civilisation [12]. Consequently, there arises a possibility of controlling progression of myopia through the modification of the environmental factors inducing this progression.

Some of the most frequently used treatments to manage progression of myopia are ortho-keratology (Ortho-K), soft bifocal and multifocal contact lenses (MFCLs), and low concentration (0.01–0.05%) atropine eye drops. Very few randomised controlled clinical trials (RCTs) to establish treatment efficacy of either Ortho-K or low concentration atropine have been published. Moreover, most RCTs of Ortho-K and soft MFCLs had high drop-out rates, which can mar the conclusions. Apart from reducing the statistical power of the comparisons, due to the smaller effective sample size, high drop-out rates may be related to discontinuation of treatment due to side-effects compromising any analysis based on intention to treat that assumes drop-outs are missing at random. According to Armijo-Olivo et al. [13], trials with missing data exceeding 20% have an increased Type I error as well as loss of power, which cannot be reliably compensated by imputation of missing data. The myopia progression intervention with the largest number of RCTs published is progressive addition spectacle lenses (PALs) (see, for example, table 1 in [14]), all with low drop-out rates <20%.

We have identified 9 RCTs on the effect of PALs on progression of school myopia in children published in the literature and added to them one quasi-randomised (alternate allocation) controlled clinical trial with well-balanced treatment and control groups with respect to the main parameters that could influence progression (the first intervention in [15]). All of the selected trials have used a constant addition power in the narrow range between +1.50 D to +2.00 for the trial duration. To compare the mean values of efficacy of PALs to one other popular intervention used for myopia control, we have also identified 9 published RCTs of soft MFCLs with constant interventions but a wider range of addition powers (+1.50 D to +2.50 D). Given the thoroughly documented search results by multiple authors for the myopia control trials up to 2020, no systematic database searches were performed, as most of the clinical trials of myopia control have already been identified in the recent IMI Interventions Report [14] and, at least in the case of PALs, are well known from the previously published systematic reviews and meta-analyses [16,17,18,19,20,21].

All of the published meta-analyses of myopia control trials used the absolute change in refraction or axial length as the main variables, which does not provide the data to calculate the percentage efficacy and might lead to higher heterogeneity of the data, if absolute efficacy varies with the baseline progression rate. In addition, some of the earlier meta-analyses have made other choices that were not helpful for evaluation of specific interventions. Li et al. [16] have mixed two different interventions (3 bifocal lens trials and 6 PAL trials) in one meta-analysis, and combined trials of different durations in one model (e.g., 18-months follow-up together with the 36-months follow-up results). As a consequence, the heterogeneity of the data set increased (the heterogeneity parameter I^2^ ≥ 65%) making the results more uncertain than they could be. Walline et al. [17] have included only three trials of multifocal spectacles (PALs) in their 12-months follow-up meta-analysis, two of which we have considered as equivocal. This group has updated its systematic review of interventions to slow down progression of myopia more recently [20] using the same methodology but has decided to include only 5 studies using PALs and 4 studies of soft bifocal and multifocal contact lenses into their meta-analyses of the multifocal lens interventions. Huang et al. [18] performed a Bayesian random-effects network meta-analysis for most possible treatments but chose to calculate the mean treatment effect per year ignoring the possible variation of the treatment with different follow-up times. There were 7 PAL trials and 3 soft MFCL studies included in their meta-analyses of multifocal lenses. Heterogeneity parameter I^2^ varied from 51% to 95% for various treatments with the lowest figure corresponding to the analysis of PALs but still representing a high level of heterogeneity and consequent uncertainty. Kaphle et al. [19] represents the most recent meta-analysis of spectacle lens trials for myopia control in children. It differs from the other studies in the time periods of comparison. They have compared the absolute progression rates over the successive 6-months or 1-year periods to gauge how long the efficacy of the intervention lasts. Since the data required for such meta-analysis are not usually published, they had to contact the authors of different studies and ask for additional unpublished data. This has limited the number of available studies, and they decided to include interventions different from PALs, such as bifocal lenses and even radial refractive gradient lenses, which are positively aspherised single vision lenses based on a different theory of myopia. Inclusion of different modalities of myopia control in different groups being analysed led to significant inconsistencies of the outcomes between those groups when comparing results over 6 M intervals to those over 12 M intervals, and high levels of heterogeneity in the latter group (I^2^ > 65%).

Our meta-analyses of PALs include the seven trials from the Huang et al. [18] PAL meta-analysis, as well as two other high-quality RCTs [22,23] and an additional 2006 study [24], but differs in two respects. We compare the relative increases in measures of myopia progression for PALs and single-vision spectacle lenses (SVLs), rather than absolute increases. We perform separate analyses for different follow-up periods, rather than assuming a pro-rata annualised effect. These differences in analysis lead to more consistent results from the RCTs, and a ready assessment of the percentage efficacy of PALs that is more precise and easier for eye care practitioners to communicate to parents and patients.

## 2. Methods

A Bayesian hierarchical model was used for the analysis of the change in spherical equivalent refraction (SER), when wearing multifocal treatment lenses, relative to the change in SER, when wearing SVLs. A similar analysis was performed for changes in axial length (AL). The treatment effect is modelled as the ratio of the change in refraction, or axial length, using multifocal lenses to the corresponding change using SVLs. Therefore, a treatment effect of 1.0 represents no benefit and a treatment effect of 0.6, for example, represents a 40% reduction in progression. For both analyses, the mean treatment effect across the hypothetical population of all possible studies is denoted by μ, and the studies in the corresponding sample, of size N, are indexed by i from 1 up to N. The underlying mean treatment effects for different studies, θ*_i_*, are supposed to vary about μ with a standard deviation of τ. This variation of study means about some overall mean is the defining feature of the random effects model. In a Bayesian analysis we model our knowledge about μ, θ*_i_* and τ by probability distributions. Our knowledge consists of prior information together with the observations from the studies.

The studies provide estimates, y*_i_*, of the θ*_i_*, and estimated values of the standard errors, σ*_i_*, which allow for sampling errors in the estimation of θ*_i_*. It is assumed that the θ*_i_* are normally distributed about μ, and that the y*_i_* are normally distributed about θ*_i_*. So, the model can be summarised by:$$\theta_{i}\;\sim \;N\left(\mu ,{\;\tau }^{2}\right)$$
$$y_{i}\;\sim \;N\left(\theta_{i},{\;\sigma }_{i}^{2}\right)$$

The observed data are the y_i_ and the associated estimated standard errors, that are taken to be the σ*_i_*. The objective is to estimate the posterior distributions for μ, θ*_i_* and τ, given the observed data and the prior information. The model does not allow for uncertainty due to estimation of σ*_i_* from the data, but the effect of this simplification on the conclusions is negligible.

The prior information about μ and τ is modelled by the following probability distributions:$$\mu \;\sim \;N\left(\mu_{0},{\;\sigma }_{0}^{2}\right)$$
and
$$\tau \;\sim \;\mathrm{HC}\left(0,{\;\tau }_{0}\right),$$
where HC is the half-Cauchy distribution with the lower bound 0 and median $\tau_{0}$, which has been recommended for priors of the variance component, as well as having other desirable properties [25]. The parameters of the prior distributions for the parameters in the model are referred to as hyperparameters. There are several options for setting values of the hyperparameters $\mu_{0}$, $\sigma_{0}$, and $\tau_{0}$. In this meta-analysis we use mildly informative priors derived from the meta-analysis of the clinical trials with bi-focal spectacle lenses—an intervention closely related to PALs and based on the same theory—reduction of accommodative lag. Furthermore, the consequences of these choices on the inferences are investigated with a sensitivity analysis. We have made use of the Bayesian random-effects meta-analysis implementation in the open source R (V3.5.2) software [26] bayesmeta [27].

The main practical advantage of a Bayesian meta-analysis is that the Bayesian approach can incorporate prior information about the parameters that are being estimated. This prior information consists of most likely values, together with an assessment of uncertainty about these values, and can include expert opinion, lower quality data than the trials considered in the meta-analysis, and data that are related to but not directly comparable with the trials in the meta-analysis. If there is no prior information the assessment of uncertainty is set to very high values. Moreover, the Bayesian meta-analysis has a nice intuitive interpretation, and advantages that the estimate of the standard deviation of a trial means about the overall mean can only be positive and the estimates of the trial means are adjusted by drawing them closer to the overall mean thereby reducing the estimation error. A Bayesian approach is also ideally suited for an adaptive meta-analysis in which results of future trials are incorporated as they become available [28]. The drawback to Bayesian analysis is that it typically requires Monte Carlo simulation from probability distributions, although this computational overhead is being eroded with modern software and computers.

The measurements of subjects are usually reported as SER in dioptres (D) and axial length of the eyeball in millimeters in both treatment and control groups. The values of these two variables may be affected by the age distribution, ethnicity, certain selection criteria (e.g., past myopia progression rate) and other characteristics of the cohort of the trial, which adds to the heterogeneity of the data. Therefore, scaled progressions (SP) have been defined for both SER and AL, and are taken as the observations y*_i_*. The SP, y, for the case of SER is defined as follows, and the definition for the case of AL is similar. Assume the mean progression of SER in the treatment group of a typical trial is v with a standard error of δv, and the mean progression of SER in the control group of the same trial is u with a standard error of δu. Then define y as follows:(1)$$y=\;\frac{v}{u}.$$

According to the rules of propagation of uncertainties, for independent random variables u and v, the standard error of the quotient is approximately:(2)$$\delta y=y\sqrt{{\left(\frac{\delta u}{u}\right)}^{2}+{\left(\frac{\delta v}{v}\right)}^{2}.}$$

Equations (1) and (2) are used to calculate the y*_i_* and the associated standard errors σ*_i_* for the trials included in the meta-analysis.

For a quick test of what effect on the outcome variability this change of variable might have, we have picked 4 RCTs of PALs [22,23,29,30] that we have judged to be of the highest quality and compiled the relevant progression data for them in Table 1.

It can be seen that the control group progression rates and the actual changes in progression between the treatment and control groups in diopters in this set of trials vary almost by a factor of 2, but the progression in the treatment group scaled by the mean progression of the control group varies by less than 5% of the mean value.

Due to the well-known weakening of the efficacy of most myopia control treatment modalities over time, we have avoided the averaging of treatment effects over diverse follow-up periods and carried out separate meta-analyses for each follow-up period, including only those trials that have reported outcomes at each of the follow-up periods: 6 M, 12 M, 18 M, 24 M and 36 M.

The prior distributions for μ and τ are based on two preliminary meta-analyses for the scaled SER variable on the set of a conceptually related intervention of bifocal spectacle lens trials at the 12 M and 24 M follow-ups, respectively. The priors derived from the 12 M follow-up meta-analysis of bifocals were used in the 6 M and 12 M meta-analyses of PAL trials, while those derived from the 24 M meta-analysis of bifocal trials were employed in the 18 M, 24 M and 36 M meta-analyses of PALs.

## 3. Data for Meta-Analysis

### 3.1. Eligibility Criteria

We have selected the relevant clinical trials of PALs based on the following criteria: (1) Study design: prospective studies with randomised parallel controls wearing SVLs; (2) Participants: 6–15 years old children; (3) Treatment: progressive addition lenses with a fixed addition power ≥ 1.50 D; (4) Outcomes: cumulative myopia progression (change in SER) and, optionally, axial elongation (change in AL) from baseline at different visits. Data for the meta-analysis was extracted independently by two reviewers (X.G. and S.V.) and cross-checked. The following information was compiled from all studies: authors, publication year, study design, age of participants, sample size, length of follow up, near addition of the PALs used, and reported outcomes (both primary and secondary, if available) at each follow-up.

### 3.2. Study Selection

The third criterion for the trial selection has led to the elimination of the intervention with the +1.00 D addition positively aspherised PAL in [23]. This trial was the only one using such a low addition power and despite its large size found no evidence that the test lens was effective in slowing down progression of myopia in 6- to 12-year-old myopic children, except for the subgroup that had no parental myopia.

The data from the trial of Leung and Brown [15] was of lower quality than the others, because non-cycloplegic subjective refractions were used to follow progression of SER, the principal investigator was unmasked, randomisation of treatment by alternate allocation relied on the order of appointments being unrelated to possible confounding factors, and small sample sizes. Moreover, there was an unplanned third group wearing +2.0 D PALs when the stock of the originally planned +1.50 D add PALs of the discontinued product being used has run out in Hong Kong. Since the +2.00 D add group was quite small (N = 14 completed the trial) compared to the target N = 40 for the test group, and the recruiters have abandoned even the quasi-randomisation of lens assignment to this group when a certain number of control group wearers was reached, we decided to exclude the data of this second treatment group from the meta-analysis.

In some trials, the standard errors of the change in SER or change in AL are given only for the last visit. In the case of trial by Yang et al. [31] we have contacted one of the authors (Weizhong Lan) who has provided us with the unadjusted mean refractions, as well as the depths of the vitreous chamber in each group and the associated standard deviations at each follow-up visit. For this trial, we have used the scaled changes in vitreous chamber depth as a substitute for the AL progression. In other cases, the standard errors for the change in SER or change in AL at intermediate visits are assumed equal to the standard errors at the last visit. This assumption has little effect on the inferences made. The standard errors of the SPs in SER and AL were derived from the published data using Equation (2).

The Hao et al. [24] trial has only been published in the abstract form without the detailed results at each follow-up. However, the bar chart of SER progression at all follow-up visits was presented in a poster session at the International Myopia Conference 11 in Singapore in 2006, a copy of which was available to us. Although it has never been included in the previous published meta-analyses, we have judged it as a trial of significant merit with a large sample size followed up over a 3-year period with the 6-monthly visits.

The data from the [24,32,33] trials were digitised from the plots presented using GetData Graph Digitiser V2.26 [33]. We have found a problem with the Edwards et al. [32] trial data in that the SER progression results used to produce figure 3a deviated significantly from those presented in table 4 of the published paper. We have contacted one of the authors of the paper to see if the discrepancy could be resolved but, due to the long passage of time since the trial was completed, we were unsuccessful in determining which values are correct. Since the original presentation by Marion Edwards of the trial results to the press relied exclusively on the graphical form (figure 3a,b published in the paper), and the table 4 was absent from the original manuscript submitted to the Investigative Ophthalmology & Visual Science Journal and must have been added later at the reviewer’s suggestion, we have decided to use the data from figure 3a in our meta-analysis. This issue did not arise in the AL progression data, as the data from figure 3b matched that presented in table 4 of the paper.

The Hasebe et al. [34] trial had a cross-over study design where the treatment lenses of the test group and control group have been exchanged after the 18-months follow-up visit. The SER progression results published in the paper were reported only for the 18-months and 36-months follow-ups. Since the trial period after 18 months did not have a proper control group that was not a subject of any prior myopia control treatment, the results of the cross-over phase of the trial were ignored. Although the follow-up visits were taking place every 6-months, the results of the 6-months and 12-months visits remained unpublished. The first author (Satoshi Hasebe) of the paper has kindly provided additional data from the other follow-ups to us after our request, and we were informed that the open-field auto-refraction measurements during the interim visits were carried out without cycloplegia to reduce stress for children taking part in the trial. Consequently, we have used the unpublished results of the first phase of this trial at all follow-up visits up to 18 months from baseline provided by the first author. We have calculated the unadjusted progression from the non-cycloplegic SER data and compared the results at 18 months to those derived from the adjusted cycloplegic auto-refraction at the same follow-up. The average retardation of progression at 18 months from the adjusted cycloplegic auto-refraction was 25%, while the corresponding value from the unadjusted non-cycloplegic auto-refraction at the same follow-up visit was 28%. The discrepancy was considered small enough to justify the use of non-cycloplegic data to present a fuller picture of the outcomes of this trial. The compiled data for the meta-analysis of PAL trials has been tabulated in the Appendix A).

### 3.3. Trial Quality Assessment

The quality of the 10 trials of PALs included in the meta-analysis was assessed using Jadad scores (Table 2), and the trials with an overall score of 4 or above were classified as high-quality trials [35].

This assessment of trial quality was based on the blinding procedures, drop-out rates, and method for randomisation. Those with Jadad scores lower than 4 were classified as “equivocal” and were excluded from the meta-analyses of the high-quality subset of the data. The meta-analyses of the SER progression component have treated the results of the trial by Edwards et al. [32] also as “equivocal” due to the ambiguity of the published results, as discussed earlier.

### 3.4. Prior Information for Meta-Analysis of Progressive Addition Spectacle Lenses (PALs)

The following publications have provided the source of data for the 5 RCTs of bifocal spectacle lenses used to generate prior distributions: [37,38,39,40,41,42]. The last of these publications by Cheng et al. presents a trial of two interventions with the +1.50 D add executive bifocal and a prismatic version of it with 3Δ base-in prism in the near segment. Due to the large heterogeneity of this data set, they could only provide mildly informative prior distributions for the main meta-analyses of trials using multifocal spectacle and contact lenses.

### 3.5. Comparison of Meta-Analysis of PALs with the Meta-Analysis of Soft Multifocal Contact Lenses (MFCLs)

To compare the main outcome of the meta-analysis for the progression of SER after 12 M follow-up to an alternative intervention of soft MFCLs, one of the reviewers (S.V.) has consulted table 2 of the recent IMI Interventions Report [14], which had 5 MFCL trials marked as “randomized” in their list of such interventions: [43,44,45,46,47] (see table 2 of their Report). After careful review of each of the original corresponding publications, it was found that two of those trials did not match the PAL trial selection criterion of the lenses having the addition power of at least +1.50 D. One of these two [45] has employed a very wide range of interventions having addition powers from +0.25 D to +3.75 D in one treatment group with no differentiation between different myopic defocus levels in the reported analysis, unlike any of the PAL trials where all children in the intervention group were wearing the same addition power. The other trial used a very low +0.50 D addition in the intervention group [43]. Moreover, we have judged the last trial from this group [47] as inadequately randomized because there were 42% more participants recruited into the treatment group than the control group. Due to the trial design, the participants could not be blinded to their intervention (contact lenses vs. spectacle lenses), which led to a much larger number of children randomly assigned into the control group dropping out early.

Consequently, we have decided to exclude these three trials from the meta-analysis and added seven interventions from three recently published trials [48,49,50], which made a total 9 RCTs of soft MFCL for meta-analysis. A serious limitation of this dataset compared to that of PALs is the high level of drop-outs, with 6 out of the 9 trials exceeding the 20% threshold at the final follow-up, and 5 of them recording >40% loss to follow-up.

## 4. Results

### 4.1. Summary of Posterior Distributions in Bayesmeta

The bayesmeta () function provides the mode, median, and mean of posterior distributions for μ and τ, but uses the median for the point estimates of μ and τ in the forest plots. Therefore, we report the median values in tables to be consistent with the forest plots. The differences between the means and medians of the posterior distributions are slight—no more than around 0.005.

The standard errors (SE) of µ and τ are given as the standard deviations of the posterior distributions. There is an approximate 0.67 probability that µ and τ lie within one SE of their means. Also, since the posterior distributions for µ are approximately normal, 95% (Bayesian) credibility intervals for µ at the follow-ups are medians plus or minus two SEs. The posterior distribution for τ is negatively skewed but approximate 67% credibility intervals for τ at follow-ups are given by medians plus or minus one SE.

The relative heterogeneity (I^2^) in the Bayesian meta-analysis has been defined as $\frac{\tau^{2}}{{\hat{\sigma }}^{2}+\tau^{2}}$, where ${\hat{\sigma }}^{2}$ is the ‘typical’ within-study variance and is calculated using Equation (9) of [51].

### 4.2. Meta-Analyses of Bifocal Lens Trials

The outcome of the meta-analyses of the five bifocal spectacle lens trials on the SER after 12 M and 24 M follow-ups are shown as forest plots in the Appendix A, respectively. The estimates of the parameters of the posterior distributions from these meta-analyses are listed in Table 3.

Based on these results, we have set up the prior distribution of μ for both the 6-months and 12-months follow-up of the scaled SER progression in PAL trials as a normal distribution with the mean $\mu_{0}=0.70$ and $\sigma_{0}=0.1$, and the prior distribution for τ as a half-Cauchy distribution with the scaling parameter $\tau_{0}=0.1$, which corresponds to a 0.5 probability that τ<0.1 and a 0.2 probability that τ>0.3. In the meta-analysis of the 18-months follow-up of SER progression in PAL trials, we have increased the three hyperparameters $\mu_{0},{\;\sigma }_{0},{\;\tau }_{0}$ to 0.75, 0.15 and 0.15, respectively, and for the 24-months and 36-months follow-up they were set to 0.75, 0.2 and 0.2, respectively.

### 4.3. Analysis of the Progression of Myopia When Wearing PALs

The primary outcome of the meta-analysis of the PAL trials was progression of myopia, defined as the scaled progression of SER. A total of 8 meta-analyses for the primary variable were calculated: 6, 12, 18, 24 and 36-month follow-ups with all trials included and 12, 24 and 36-months with the 4 equivocal trials excluded. The associated estimates of the parameters μ and τ and their standard errors are presented in Table 4. A 95% (Bayesian) credibility interval for µ at the 12-months follow-up is [0.653, 0.785], and the corresponding 67% credibility interval for τ is [0.007, 0.067].

The graphical representations of the selection of meta-analyses of the SER progression at 12 and 24-months follow-ups with all trials included are shown as funnel plots in Figure 1, and as forest plots in Figure 2 and Figure 3. The same representations at the remaining follow-ups can be found in the Appendix A.

There was an increase in refractive errors of subjects in each trial at different follow-ups; the increase in the absolute value of SER of subjects wearing SVLs tended to be greater than that of subjects with PALs. The mean scaled progressions of SER when wearing PALs relative to SVLs can be readily converted to percentage efficacies of the intervention by subtracting the value of SP from 1 and multiplying by 100%. This conversion provided the mean percentage efficacies at the 6-months, 12-months, 18-months, 24-months and 36-months follow-ups, which were 27%, 28%, 20%, 20% and 15%, respectively, with all trials included. When we excluded the 4 equivocal trials, the mean percentage efficacies were 30%, 20% and 19% at 12-, 24- and 36-month follow-ups, respectively. These results imply that the percentage of retardation of myopia progression with progressive lenses was approximately constant for the first 12 months (6 M and 12 M) of follow-up, and then it has decreased beyond that time.

The results of the calculation of the relative heterogeneity I^2^ for the primary variable SER are shown in the last row of Table 4. They reveal low rates of heterogeneity of around 10% for the first 12 months of follow-up, and still <25% for the 24-months follow-up with the equivocal trials excluded. The only high value of >50% relative heterogeneity has been found at the 36-months follow-up, with the 1 equivocal trial excluded, but this analysis was limited because there were only two good quality trials available at this long follow-up.

### 4.4. Analysis of Axial Elongation with PALs

The secondary outcome measure was the scaled progression of AL of the eye. One of the equivocal trials [24] did not report any AL measurement data, while another [36] only reported the axial length progression at the 18-months follow-up. As was mentioned earlier, the AL measurements of [32] were not ambiguous, and there was no known reason to mark them as “equivocal”. Consequently, there was only one trial for the secondary variable of AL in our set that could be classified as equivocal [15]. There were only small differences in the estimates of μ and τ from separate meta-analyses excluding the one equivocal trial, so we only present results for the secondary variable for all trials.

We did not have a sufficient amount of AL data in the bifocal lens trial data set to establish an informative prior distribution for this variable. Therefore, we have adjusted the hyperparameters derived from the SER data to correspond to the expected lower efficacy and greater heterogeneity of the AL progression data. The choices we have made for the prior distribution of μ for both the 6-months and 12-months follow-up of the AL progression in PAL trials were a normal distribution with the mean $\mu_{0}=0.80$ and $\sigma_{0}=0.2$, while that of τ was set up as the half-Cauchy distribution with the scaling parameter $\tau_{0}=0.1$. In the meta-analysis of the 18- and 24-months follow-up of AL progression in PAL trials, we have increased the hyperprior $\tau_{0}$ to 0.2 keeping the other two values the same.

The funnel and forest plots had been generated at 6, 12, 18 and 24-months follow-ups for all trials which have collected the axial length progression data. The funnel plots for the 12- and 24-months follow-ups are shown in Figure 4, while the corresponding forest plots are displayed in Figure 5 and Figure 6. The corresponding plots for the meta-analyses at 6 and 18 months are displayed in Appendix A. The estimates of the µ and τ for the scaled progression in AL, and their standard errors together are displayed in Table 5.

Similar to the change in SER, there was an increase in AL of subjects in each trial at different follow-ups. PALs did appear to retard axial elongation at all follow-up visits in a statistically significant way. However, the magnitude of the retardation of axial elongation was only around 0.6 of that of the retardation of the SER progression, and there did not appear to be any clear tendency for the effect to weaken over time. The calculated values of the relative heterogeneity for the axial length progression appear to be higher than those for the SER progression. They are also shown in Table 5.

It is not clear why the axial length measurement outcomes are less consistent than those of (mostly) cycloplegic autorefraction. It could be due to the two different techniques of measuring axial length of the eyeball being used by different trials: ultrasound A-scan and interferometry, respectively.

### 4.5. Sensitivity Analysis

The sensitivity analysis to gauge the impact of the variations of the priors on the estimates of μ and τ tested for the 12-months follow-up analysis has been run with the values of $\mu_{0}$ being 0.6, 0.7 and 0.8, those of $\sigma_{0}$ taking the values of 0.1 and 0.2, while the $\tau_{0}$ values were 0.1, 0.2 and 0.3. The results with different choices of prior distributions can be found in Appendix A.

As can be seen from the second column of Appendix A when all trials are included, the maximum variation of μ from the estimate of 0.719, given in Table 4 and Figure 2, was 0.012. The corresponding estimate of μ was 0.731, which occurred with a prior estimate of μ of 0.80. To put this in context, the variation is about one third of the standard error of μ. When the 4 equivocal trials were excluded, the estimate of μ given in Table 4 is 0.699 and the maximum deviation was 0.018, which is less than half the standard error of μ.

### 4.6. Meta-Analysis of Soft MFCLs Trials for a Comparison

Comparison of the results of our meta-analysis of PALs to control progression of myopia with other interventions for myopia control is difficult, as reliable clinical data from RCTs for other modalities of myopia control is scarce. The only possible exception could be the multifocal contact lens interventions, 9 RCTs of which we selected earlier. The progression data for these 9 interventions with such soft contact lenses has been extracted and compiled for the 12 months of follow-up only, as the majority of those trials had high drop-out rates. They include data for a total of 982 children having the mean age at baseline of 10.3 years and mean SER of −2.4 D, which are comparable to the mean values of the PAL trials cohort. A Bayesian meta-analysis hierarchical model for the 12-months follow-up scaled SER progression data in this set of 9 RCTs, using the methodology and prior distribution applied to the PAL intervention models at the same follow-up, was fitted. The model suggests a moderate level of heterogeneity in the data with $I^{2}=36\%$. The funnel plot of the data set for the SER progression is shown in Figure 7, which reveals that the source of the heterogeneity are the two outliers outside the 95% credibility interval having the scaled progression of SER close to 0.4, which correspond to the trials of the Cooper Vision MiSight™ bifocal contact lens [48] and the high addition (+2.50 D) Cooper Vision Biofinity^®^ MFCL [50]. Since the MiSight™ lens is also reported to have the high +2.50 D addition [52], and it has been recently found that the lower +1.50 D addition of the Biofinity^®^ MFCL has a considerably lower efficacy, it would be tempting to conclude that these two outliers represent a different kind of intervention from that used in the PAL trials, which have tested addition powers ≤2.00 D. However, the remaining 7 MFCL interventions in the model appear to show a moderate 24% mean retardation of SER after 12-months follow-up with very low heterogeneity (8%) despite a large variety of different lens designs and addition powers, including two trials with the +2.50 D addition—the Defocus Incorporated Multiple Segments (DISC) trial [44] and the Test 1 intervention in [49], being used. On the other hand, these two trials of other +2.50 D add soft contact lenses were affected by very high (>40%) drop-out rates, unlike the two Cooper Vision lenses trials. This may indicate that the test lenses were poorly tolerated and retained wearers’ compliance with the treatment regime could have also been affected. Indeed, the DISC trial [44] reported serious deviations from the full time wear instructions and in subgroup analyses has demonstrated higher efficacies of the test lens when the reported wear times were closer to full-time wear. So, it is possible that there is a significant jump in efficacy of soft bifocal contact lenses when the addition power is increased to +2.50 D and they represent a qualitatively different intervention to their lower addition counterparts. Figure 8 displays the forest plot for this model having the mean value of scaled progression difference of 0.709 [95% credibility interval (CI): 0.627, 0.787], which corresponds to the mean relative efficacy of the CL intervention being 29% [95%CI: 21, 37].

## 5. Discussion

The main outcome of our meta-analyses was that PALs have slowed down progression of myopia at every follow-up up to 3 years in a statistically significant way. The effect size was constant for the first 12 M but tended to weaken after that. This contradicts the finding of Kaphle et al. [19] that the effect size is more than halved in the second 6 months compared to the first. The estimates of overall scaled progression of SER were not substantially affected by the removal of the equivocal trials, although the associated standard errors increased slightly. PALs have slowed down progression of myopia in the first 12 months by about 28% on average, but their efficacy appears to show a tendency to weaken after that. The estimated drop in efficacy between 12 months and 24 months is 0.08 of SP and between 24 months and 36 months it is 0.05. So, there appears to be a weakening in efficacy over time. However, this weakening has not been determined with substantial precision, and 95% credibility intervals for differences between 12 months and 24 months, 24 months and 36 months, and 12 months and 36 months all include 0.

The secondary outcome is the overall scaled progression of AL at 6-, 12-, 18-, 24-months follow-ups. Progressive lenses slowed down the increase in axial length in the first 12 months by about 18%. The effect size is lower compared to that of the refraction, which is not surprising, as, in children of the age range included, some of the axial elongation is being compensated by the crystalline lens thinning [53,54] and does not contribute to the progression of myopia (Satoshi Hasebe, personal communication). Assuming the intervention does not affect the crystalline lens thinning, the relative values of the change in axial elongation would be expected to be lower than the changes in the relative values of refraction in such population because the denominator of the relative progression of SER is compensated by the thinning of crystalline lens, while that of the relative progression of AL is not compensated, i.e., the denominator is relatively larger for AL making the ratio smaller.

To estimate the expected reduction in the change of the relative axial elongation compared to the relative SER progression, we will use the classical rule of thumb derived from schematic eye models that for every 1 mm of axial elongation the refraction changes by −3 D, if there is no change in either the corneal curvature or the crystalline lens thickness. As an example, let us assume that the axial elongation of the eyeball is 0.75 mm and that the 0.25 mm (one third) out of those 0.75 mm is compensated by the crystalline lens thinning, as was seen in the Correction of Myopia Evaluation Trial (COMET) trial [55]. Therefore, myopia causing axial elongation in such a hypothetical trial would be 0.75 − 0.25 = 0.50 mm, which corresponds to approximately −1.5 D of SER progression, according to the classical rule of thumb. If PALs reduce the axial elongation by 0.15 mm, say, which would correspond to the reduction in SER progression of 3 × 0.15 D = 0.45 D, the treatment effect on refraction is 0.45D/1.5D = 30%, but the relative effect on AL progression will be 0.15 mm/0.75 mm = 20%. The obtained results for the ratio of the relative efficacy of the treatment of SER to that of the AL after 12-months follow-up (0.281/0.184 = 1.53) and the 24-months follow-up (0.199/0.140 = 1.42) are in excellent agreement with the expected ratio of 1.5 from the above assessment of the role of crystalline lens thinning to partially compensate axial elongation. This is sometimes not the case in trials of certain other myopia control interventions, where relative efficacies close to 1:1 ratio for the SER and AL have been reported [48,56]. That is quite puzzling, especially when the ratios of progression of SER and AL in the control groups of those trials show conversion factors closer to 2 D/mm than 3 D/mm (2.11 and 1.76 D/mm at the final visits, respectively), indicating a significant effect of crystalline lens thinning on the slower progression of myopia than would be expected from the eyeball elongation alone.

We believe that PALs are currently the only modality of myopia control intervention that have a sufficient number of published good quality clinical trials with a consistent treatment effect to make a meta-analysis of their outcomes meaningful. The presence of a subset of 3 trials run in the USA with a predominant Caucasian cohort of subjects and the overall low level of heterogeneity in trial outcomes for the primary variable of scaled SER progression in the first 12–18 months of follow-up does not suggest significant variation in relative treatment efficacy across the races or ethnicities.

One limitation of this meta-analysis is that it provides estimates of the efficacy of the treatment in the cohort of children with a mean age of 10.0 years old and an average refraction of −2.6 D. At least one of the PAL trials in our set [23] has found that the percentage efficacy in a subgroup of younger children (6 – 9 years old) in the first 12 months was almost twice as high as in the whole group (57% vs. 30%). Consequently, the mean efficacy percentages are of limited use in predicting the efficacy for each individual patient. They just establish a benchmark against which other interventions can be assessed, if the mean age and baseline refraction of the cohorts are comparable.

Comparing the estimates of the mean efficacy of the soft MFCL based on the meta-analysis presented in the previous section to those derived from the PAL model of the primary variable SER at the same 12 M follow-up, it is clear from the overlap of the credibility intervals that the mean efficacies of retarding progression of myopia by the soft MFCL are not significantly different from those of PALs in the first year of intervention.

This result may suggest that the mechanism of their action is the same. Of the two prevailing theories how the myopigenic stimulus can be inhibited, the reduction of accommodative lag hypothesis appears to be more consistent with this outcome than the compensation of peripheral hyperopic shift theory. Most PALs provide a compensation for the relative peripheral hyperopic shift only in the lower field of view (superior retina), while soft MFCLs send this signal to the entire retina. We hypothesise that a slight trend to a lower mean efficacy of soft MFCLs in our meta-analysis, when the two outliers are excluded, may be due to a lower wearer compliance with this intervention compared to the spectacle lenses [44].

In order for the PAL intervention to make a significant impact on preventing children from becoming high myopes, it is imperative to understand the reasons behind the tendency for the PAL effect to weaken beyond the first year of treatment and explore the options for extending the first-year efficacy. Accommodative complacency—a longer term (~12 months) erosion of accommodative gains from plus power may be the cause. Negatively aspherised soft contact lenses have been found to reduce accommodative lag over the first 6 months of wear but at 12 months the effect has disappeared [57]. It should be noted that the near vision zone of all commercially available PALs is also negatively aspherised. In a collaborative project with the Queensland University of Technology (QUT) we are currently investigating changes to accommodative lag after 12 months’ wearing PALs by young adult myopes. The preliminary analysis suggests that such accommodative complacency does develop over time when viewing objects through the near zone of a PAL having a +1.50 D addition at the closest object distances of 25 cm, which are often used with small electronic devices [58]. Those preliminary results also indicate that an 0.5 D increase of addition power is able to restore the accommodative lag to the lower original value after 12 months’ wearing +1.50 D addition PALs for this short wearing distance.

At least one longer term (5 year) trial of Ortho-K intervention for myopia control in children [59] has shown the maintenance of a statistically significant annual retardation of progression of axial elongation over the first 3 years of the trial. Unlike the PAL trials, the Ortho-K intervention used in this and other myopia control clinical trials was variable over time. The design of the contact lenses used to aspherise the cornea was changed over the course of the trial to increase the asphericity of the cornea, as the patient’s myopia progressed. This effectively provided a higher addition power by the aspherised cornea for wearers having a higher myopia and ensured an increase of the addition power for all wearers progressing in their myopia sufficiently to record a change in the visual acuity by more than 0.30 logMAR units, as the trial progressed. We hypothesise that a similar increase in the addition power of a PAL, perhaps on an annual basis, may also extend the higher efficacy of this modality of myopia control observed over the first 12 months of the follow-up over a longer period of time, but this, to the best of our knowledge, has never been tested in a clinical trial.

## 6. Conclusions

There is convincing evidence that the rate of myopia progression is related to age and possibly to existing level of myopia (e.g., Ref. [60]). Therefore, the use of progressive spectacle lenses having a moderate range of addition powers for myopia control in children and juveniles would only be moderately effective in the first year if applied for the first time to a 10-year-old who has already passed the half-way point on their march to high myopia (−5.0 D). Soft multifocal contact lenses, which appear to have a very similar mean efficacy in the first year to that of PALs, when applied to at the same age and level of myopia, may have considerably higher efficacies when the higher addition powers are dispensed (e.g., ≥2.50 D).

Since both types of multifocal lenses analysed appear to have very similar mean efficacies and are likely to have the same or very similar mechanism of efficacy, we would recommend to eyecare practitioners the inclusion of PALs with higher addition powers than have been used routinely to date. This could be done starting with the +2.50 D addition immediately when the development of myopia has been discovered, or gradually by starting with the moderate addition even a little before myopia has actually developed, when the risk factors for developing myopia are clearly visible, and increasing the addition power every year to maintain PAL efficacy in controlling progression of myopia over more years.

Our experience with adaptation to progressive lens wear among presbyopes suggests that it is much easier to accept those lenses when people start wearing them in lower addition powers and gradually transition to the higher additions. We would expect that a gradual ramping up in addition power over the years will promote compliance with PAL wear in younger subjects as well. Based on the preliminary results of the ongoing collaborative research project on accommodation with the QUT, we would recommend that the addition power be increased by 0.5 D every 12 months.

## Figures and Tables

**Figure 1 jcm-10-00730-f001:**
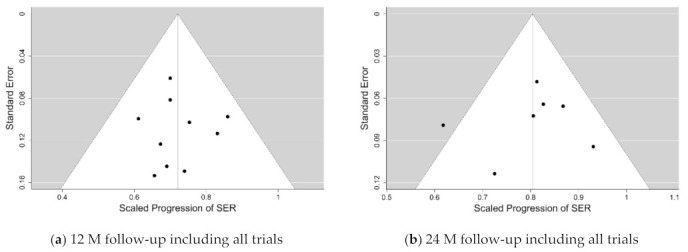
Funnel plots for the primary variable in progressive spectacle lens meta-analysis—the scaled progression of sphere equivalent refraction (SER) at the 12-months and 24-months follow-ups with all trials included (**a**,**b**), respectively). The vertical line at the centre marks the mean effect size of all trials, the sides of the funnel mark the 95% credibility interval, and the dots correspond to individual intervention outcomes.

**Figure 2 jcm-10-00730-f002:**
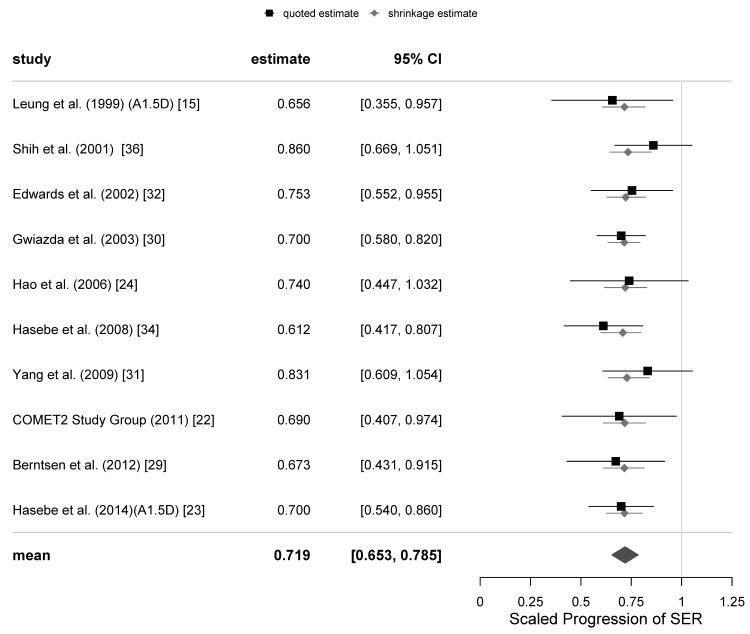
Forest plot for the scaled progression of sphere equivalent refraction (SER) at 12-months follow-up with all the trials included. The first column of the forest plots references the source of the data, the second column gives the mean effect size obtained in the study expressed as a scaled progression, the third column gives the 95% credibility interval associated with the mean effect size. The last column shows the estimate of the mean effect size and the associated credibility intervals for the mean effect of each trial (black squares and lines) and the shrinkage estimates (grey diamonds and lines), which are corresponding estimates adjusted relative to μ and τ. The centre of the large diamond in the final row corresponds to the estimated mean effect size of all the trials included in the meta-analysis, and its width corresponds to the calculated 95% credibility interval.

**Figure 3 jcm-10-00730-f003:**
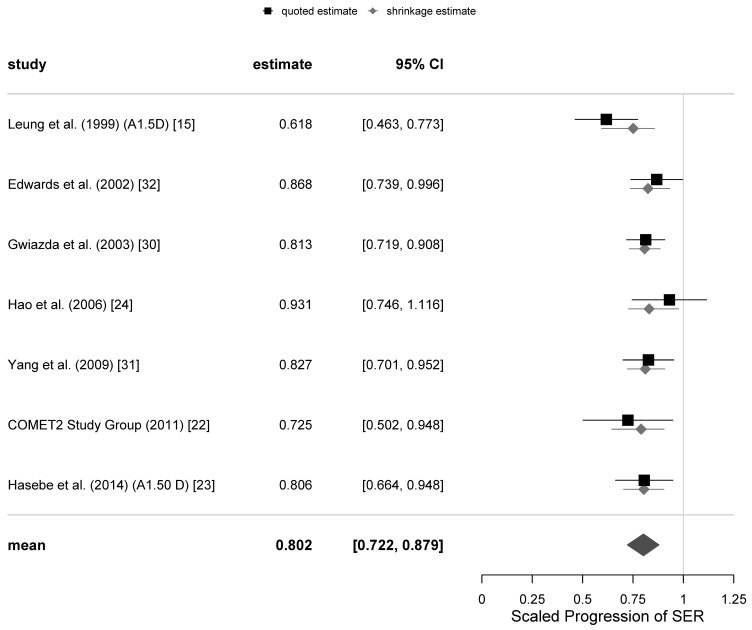
Forest plot for the scaled progression of sphere equivalent refraction (SER) at 24-months follow-up with all the trials included.

**Figure 4 jcm-10-00730-f004:**
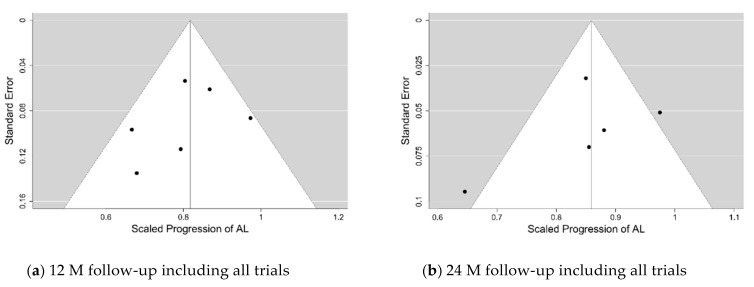
Funnel plots for the secondary variable—the scaled progression of axial length (AL) at the 12-months and 24-months follow-ups with all trials included (**a**,**b**, respectively).

**Figure 5 jcm-10-00730-f005:**
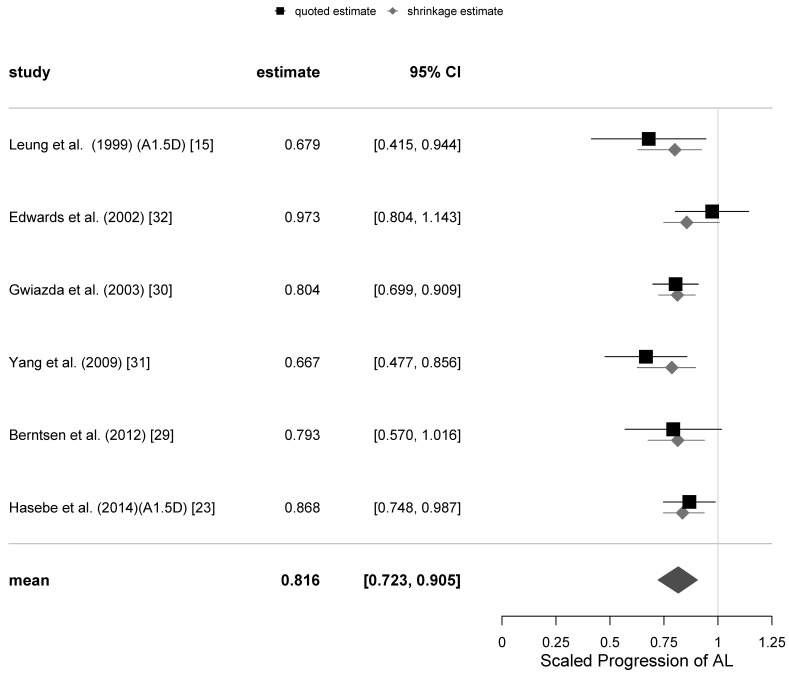
Forest plot for the scaled progression of axial length (AL) at 12-months follow-up with all trials included.

**Figure 6 jcm-10-00730-f006:**
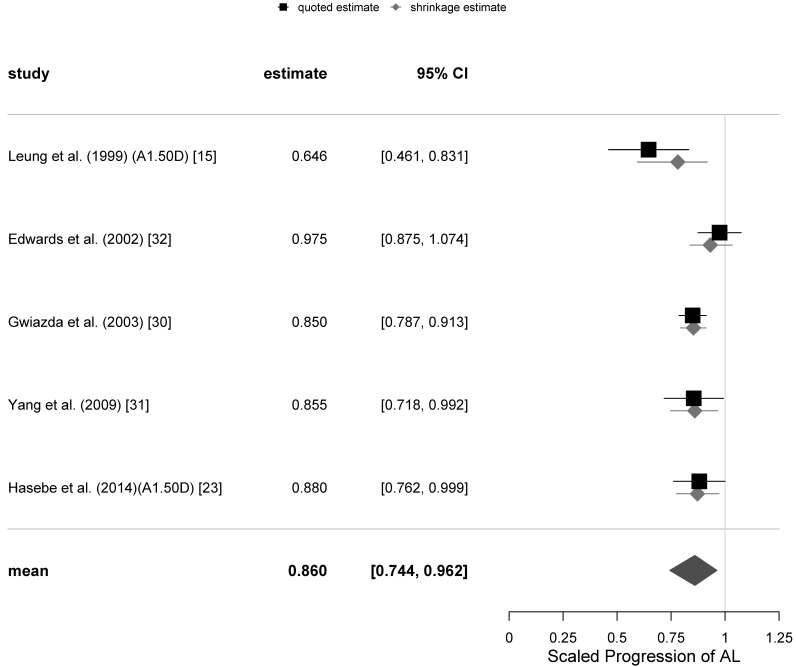
Forest plot for the scaled progression in axial length (AL) at the 24-months follow-up.

**Figure 7 jcm-10-00730-f007:**
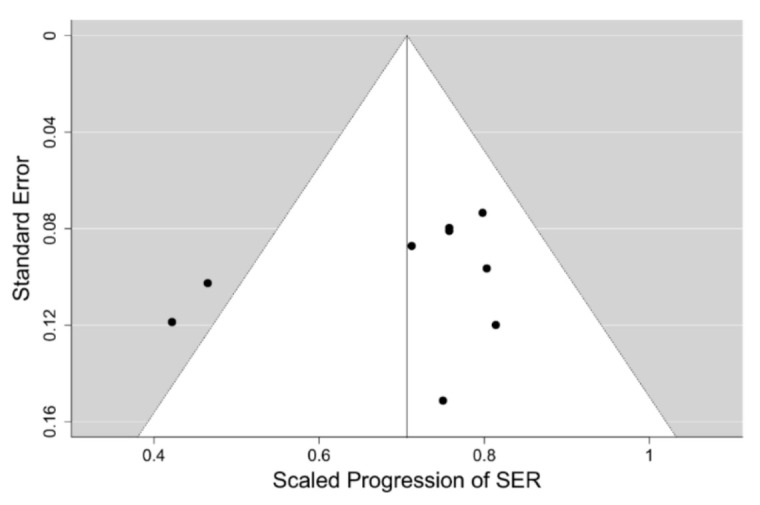
Funnel plot for the scaled progression of sphere equivalent refraction (SER) at the 12-months follow-up in the model for the 9 interventions using soft multifocal contact lenses to control progression of myopia.

**Figure 8 jcm-10-00730-f008:**
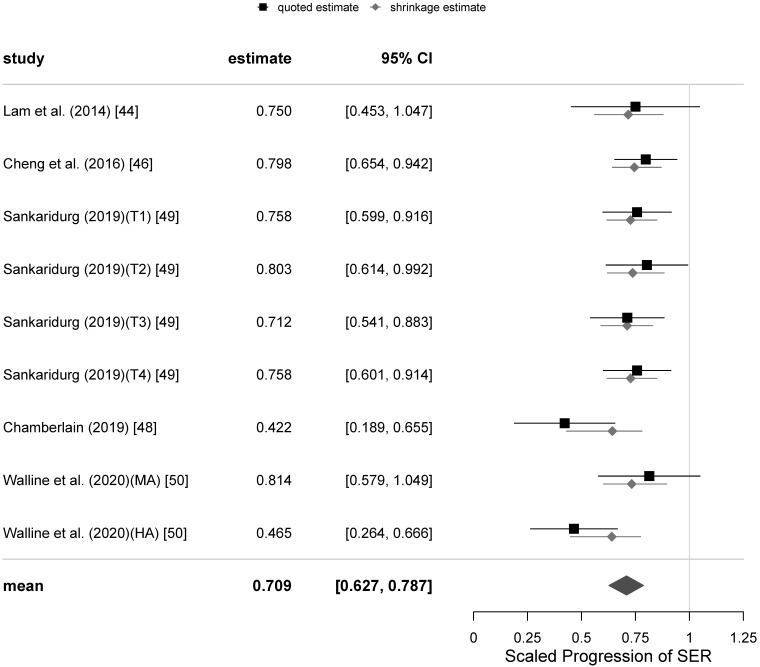
Forest plot for the scaled progression of sphere equivalent refraction (SER) at the 12-months follow-up in the set of 9 interventions using soft multifocal contact lenses.

**Table 1 jcm-10-00730-t001:** Comparison of variability of spherical equivalent refraction (SER) progression outcomes after 12 M follow-up in a subset of four highest quality clinical trials of progressive addition spectacle lenses (PALs) defined as dimensionless scaled progression to the change in progression in diopters.

Trial	Mean Age (Years)	PAL Group Progression (D)	SVL Group Progression (D)	Change in Progression (D)	Scaled Progression
COMET2 (2011) [22]	10.1	−0.29	−0.42	0.13	0.69
Berntsen et al. (2012) [29]	9.9	−0.35	−0.52	0.18	0.67
Gwiazda et al. (2003) [30]	9.3	−0.42	−0.60	0.18	0.70
Hasebe et al. (2014) [23]	10.3	−0.56	−0.80	0.24	0.70

**Table 2 jcm-10-00730-t002:** Quality assessment of the ten progressive addition spectacle lens trials included in the meta-analysis. SB—single blind; NS—not specified; DB—double blind.

Study (Year)	Randomisation	Blinding	Drop-Outs	Allocation Concealment	Jadad Score
Leung & Brown (1999) [15]	Adequate	SB	Adequate	Inadequate	1
Shih et al. (2001) [36]	Adequate	DB	Adequate	Adequate	3
Edwards et al. (2002) [32]	Adequate	DB	Adequate	Adequate	4
Gwiazda et al. (2003) [30]	Adequate	DB	Adequate	Adequate	5
Hao et al. (2006) [24]	Adequate	NS	Inadequate	NS	1
Hasebe et al. (2008) [34]	Adequate	DB	Adequate	Adequate	4
Yang et al. (2009) [31]	Adequate	DB	Adequate	Adequate	4
COMET2 (2011) [22]	Adequate	DB	Adequate	Adequate	5
Berntsen et al. (2012) [29]	Adequate	DB	Adequate	Adequate	5
Hasebe et al. (2014) [23]	Adequate	DB	Adequate	Adequate	5

**Table 3 jcm-10-00730-t003:** Estimates of model parameters derived from the meta-analyses of 5 randomised clinical trials using bifocal spectacle lenses.

Estimated Parameters	12 M Follow-Up	24 M Follow-Up
μ (SE)	0.705 (0.109)	0.749 (0.156)
τ (SE)	0.170 (0.108)	0.269 (0.151)

**Table 4 jcm-10-00730-t004:** The number of included trials, estimated values of µ, τ and their associated standard errors, the probability that µ is smaller than 1, and the values of relative heterogeneity at different time intervals with and without the 4 equivocal trials for the primary variable of the scaled sphere equivalent refraction progression.

	All Trials Included					Equivocal Trials Excluded		
Follow-Up	6 M	12 M	18 M	24 M	36 M	12 M	24 M	36 M
Included Trials	7	10	7	7	3	6	4	2
µ (SE)	0.732 (0.059)	0.719 (0.033)	0.802 (0.037)	0.801 (0.039)	0.849 (0.070)	0.699 (0.040)	0.802 (0.048)	0.813 (0.096)
τ (SE)	0.059 (0.051)	0.037 (0.030)	0.040 (0.034)	0.063 (0.048)	0.086 (0.091)	0.042 (0.037)	0.052 (0.054)	0.126 (0.155)
P (µ *<* 1)	1.000	1.000	1.000	1.000	0.985	1.000	1.000	0.975
I^2^	0.06	0.08	0.11	0.36	0.43	0.11	0.23	0.51

**Table 5 jcm-10-00730-t005:** The number of included trials, the estimated values of µ, τ and their associated standard errors, the probability that µ is less than 1, and the values of relative heterogeneity for the secondary variable (scaled axial length) at different time intervals with all the trials included.

Follow-Up	6-Months	12-Months	18-Months	24-Months
Included Trials	4	6	5	5
µ (SE)	0.866 (0.072)	0.816 (0.046)	0.885 (0.054)	0.860 (0.053)
τ (SE)	0.048 (0.064)	0.052 (0.050)	0.063 (0.065)	0.081 (0.067)
P (µ *<* 1)	0.972	1.000	0.986	0.996
I^2^	0.11	0.29	0.41	0.68

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
