# Peer review of "Bayesian Meta-Analysis of Myopia Control with Multifocal Lenses"

_jcm, 2021, doi:10.3390/jcm10040730_

Round 1
Reviewer 1 Report
General comments:
The authors propose to conduct a Bayesian meta-analysis on the effect of multifocal lenses on myopic progression in children. In particular, they aim to investigate the efficacy of PALs and multifocal soft contact lenses on retardation of myopic development. Several issues should be addressed before consideration for publication.
- Methods and Discussion sections should be significantly shortened and more focused. Meticulousediting of the manuscript is warranted to make it shorter, clearer, and improve the flow of arguments. An in-depth analysis of useful findings is needed.
- PALs have been already investigated by a number of studies, and their efficacy in retarding myopic progression has been judged as clinically modest. This review aimed at conducting an analysis examining the efficacy of PALs and multifocal soft contact lenses on slowing myopic development; however, unlike the network meta-analysis by Huang et al. (Huang J, Wen D, Wang Q, et al (2016) Efficacy comparison of 16 interventions for myopia control in children: A network meta-analysis. Ophthalmology 123:697–708.), it ignores other proven effective modalities including atropine drops and orthokeratology, for the sake of a claimed larger sample size for PALs. It appears unclear whether the authors finally propose that their findings should change current clinical practice, or could have any meaningful interpretation.
Specific comments:
Abstract:
- Page 1, lines 12-14: authors should preferably rephrase this sentence and, instead, clearly state the purpose/research question pertaining to this meta-analysis.
- Page 1, line 16: change to ‘meta-analysis’.
- Page 1, lines 17-18: please provide the exact value of included children in the analyses.
- Page 1, lines 23: ‘not statistically different’ could authors be more specific by providing a numeric value?
- Page 1, lines 24: how is ‘moderately’ defined?
- Page 1, lines 22-26: results regarding axial elongation are not mentioned in the abstract, but are later on reported in the manuscript.
Introduction:
- Page 1, line 44: These interventions are implemented not because of popularity, but due to their proven efficacy. Please rephrase.
- Page 2, lines 48-49: What were the drop-out rates for RCTs examining Ortho-K and multifocal lenses? Please provide numerical values and corresponding references.
- Page 2, lines 63-65: “To compare..+2.50D)” unclear phrasing.
- Page 2, lines 69-70: Authors acknowledge that RCTs for PALs have already been analyzed in previous studies. The following study should also be referenced: Prousali E, Haidich A, Fontalis A, et al (2019) Efficacy and safety of interventions to control myopia progression in children : an overview of systematic reviews and meta-analyses. BMC Ophthalmol 19:106
- Page 2, lines 74-76: Authors appear to comment on their findings in the Introduction section. Please include all findings under ‘Results’.
- Page 3, lines 108-116: Authors mention that they aim to analyze RCTs on PALs, which have already been included in previous published meta-analyses. The authors need to clearly explain how their analysis adds information to those already presented and convince the reader that this meta-analysis will provide benefit over what is written in the literature, ie. the recently published meta-analysis by Kaphle et al. (Kaphle D, Atchison DA, Schmid KL (2020) Multifocal spectacles in childhood myopia: Are treatment effects maintained? A systematic review and meta-analysis. Surv Ophthalmol 65:239–49.). Of note, in this paragraph there is no reference to multifocal soft contact lenses, which are supposed to be included in the analysis.
Methods:
- Page 3, line 117: change to ‘Methods’.
- Page 4, line 188: change to ‘Eligibity Criteria’.
- Page 5, line 204: remove ‘somewhat’.
- Page 5, line 213: remove ‘have’.
- Page 5, lines 219-220: the authors should explain why vitreous chamber depth has been used as an equivalent of axial length and provide a relevant reference.
- Page 4-6, lines 199-264: this part should be added as a separate sub-section named ‘study selection’.
- Page 6, lines 266-276: quality assessment should be added as a separate sub-section.
- The authors provide no information regarding the search strategy, if implemented, for this review. Under the ‘Introduction’ section (Page 2, lines 66-70), authors mention that no systematic search was performed. This would be rather inappropriate for a high-quality study; authors should clearly mention in the text on which literature search stategy they have based their findings.
- The whole ‘Methods’ section needs shortening and statistical analysis should be preferably moved after ‘quality assessment’ sub-section. A separate, clear sub-section on outcome measures should also be added.
Discussion:
- Page 15, line 474: do the authors wish to comment on a proposed mechanism for how part of axial elongation might not contribute to myopic progression?
- Page 16, lines 509-510: remove ‘around’ and provide the exact mean values.
- Page 16, lines 518-544 and figures 7-8: results should be reported under the ‘Results’ section, rather than ‘Discussion’.
- Page 18, lines 573-577: what do authors mean by ‘variable’?
- ‘Discussion’ seems to lack a clear appraisal of the authors’ findings and discussion of their clinical significance, specifically with regard to establishing guidelines for myopic progression control. In general, ‘Discussion’ appears wordy and confusing, which would not help the readers absorb any meaningful conclusion.
Author Response
Dear Reviewer 1,
Please find the responses to your comments on our manuscript attached.
Yours sincerely,
Saulius Varnas
-------------------------------------------------
Reviewer 1
General comments:
The authors propose to conduct a Bayesian meta-analysis on the effect of multifocal lenses on myopic progression in children. In particular, they aim to investigate the efficacy of PALs and multifocal soft contact lenses on retardation of myopic development. Several issues should be addressed before consideration for publication.
Actually, the paper is focussed on the Bayesian meta-analysis of myopia control intervention of multifocal spectacle lenses (PALs) as the main target of our study. The meta-analysis of soft multifocal contact lens (MFCL) intervention is only included for a single follow-up of 12M for comparison to the main analysis. The PRISMA Guidelines recommend that a comparison with other evidence be included in the Conclusion section of published meta-analyses. Since such comparison is not possible with any of the published meta-analyses in the literature because of the use of scaled variables to measure outcome in our meta-analysis, we needed to run our own meta-analysis of another intervention to enable comparison. All of the published meta-analyses of myopia control interventions are using the absolute value of change in progression between the test group and control group of either sphere equivalent refraction (SER) or axial length (AL) as the main variable.
Methods and Discussion sections should be significantly shortened and more focused. Meticulous editing of the manuscript is warranted to make it shorter, clearer, and improve the flow of arguments. An in-depth analysis of useful findings is needed.
We would like to point out that Reviewer 2 has asked to expand the Methods section to provide more detail on the advantages and limitations of Bayesian meta-analysis. If the Methods section is shortened, this would force us to skim on the basics of Bayesian meta-analysis, which, to our knowledge, has been rarely used in ophthalmology papers, and myopia control intervention meta-analyses in particular. The Huang et al. (2016) network meta-analysis is the only published meta-analysis of myopia control using the Bayesian approach known to us, but it is focussed on the network meta-analysis variant of it (head to head comparisons of the outcomes of some individual Bayesian meta-analyses of a few small clusters of clinical trials for different interventions but mostly just individual trials). Unfortunately, the methodology part of that paper is lacking in detail of Bayesian meta-analysis employed. For example, it doesn’t even mention what prior distribution was used in every meta-analysis they have conducted. We have tried to remove some parts of the Discussion section that are not essential and moved the presentation of the results of the of the meta-analysis of the soft MFCL to the Results section to address the concern with the length of this section.
PALs have been already investigated by a number of studies, and their efficacy in retarding myopic progression has been judged as clinically modest. This review aimed at conducting an analysis examining the efficacy of PALs and multifocal soft contact lenses on slowing myopic development; however, unlike the network meta-analysis by Huang et al. (Huang J, Wen D, Wang Q, et al (2016) Efficacy comparison of 16 interventions for myopia control in children: A network meta-analysis. Ophthalmology 123:697–708.), it ignores other proven effective modalities including atropine drops and orthokeratology, for the sake of a claimed larger sample size for PALs. It appears unclear whether the authors finally propose that their findings should change current clinical practice, or could have any meaningful interpretation.
We have implemented a pairwise meta-analysis using Bayesian random effects models, which can synthesise data from clinical trials limited to two treatment groups – PALs with a narrow range of addition powers compared to single vision spectacle lenses. The network meta-analysis approach used by Huang et al. (2016) is an extension of Bayesian meta-analysis implemented in different software from that they have used for the Bayesian meta-analysis (STATA version 10 and WinBUGS version 4, respectively).
We have added the Conclusions section at the end to spell out they key findings and current clinical practice change recommendations arising from this analysis.
Specific comments:
Abstract:
- Page 1, lines 12-14: authors should preferably rephrase this sentence and, instead, clearly state the purpose/research question pertaining to this meta-analysis.
Replaced the original: “Meta-analyses of juvenile myopia control interventions published to date do not provide reliable guidelines for the mean percentage efficacy in slowing down progression of myopia by a specific intervention over defined time periods, derived from Randomised Controlled clinical Trials (RCTs) with consistent outcomes” with “The aim of this study is to provide reliable guidelines for the mean percentage efficacy together with the 95% credibility interval in slowing down progression of myopia by a specific intervention over defined time periods, derived from a substantial number of Randomised Controlled clinical Trials (RCTs) with consistent outcomes.”
- Page 1, line 16: change to ‘meta-analysis’.
We have used the plural form because we have presented the outcomes of a total of 13 meta-analyses: 8 meta-analyses of PALs for the scaled SER variable at different follow-up times ranging from 6M to 36M with all trials included and with equivocal trials excluded, 4 meta-analyses of PALs for the scaled AL variable at different follow-ups ranging from 6M to 24M, and one meta-analysis of soft multifocal contact lenses at 12M follow-up for the scaled SER variable.
- Page 1, lines 17-18: please provide the exact value of included children in the analyses.
We have now provided the exact number of children enrolled at baseline in all of the trials of PALs and soft MFCLs, as suggested.
- Page 1, lines 23: ‘not statistically different’ could authors be more specific by providing a numeric value?
We have now provided the 95% credibility intervals for the mean values derived from the meta-analyses of both interventions – PALs and soft MFCLs – after 12M follow-up. These demonstrate convincingly the massive overlap of those 95% credibility intervals. Unlike the frequentist meta-analysis, Bayesian version of it is not based on formulating a null hypothesis and finding a p-value of it being true.
- Page 1, lines 24: how is ‘moderately’ defined?
We think that around 30% is a reasonable figure to call it “moderate”. For example, the following qualitative scale could be proposed for percentage efficacy of myopia control: very high (>75%), high (50% to 74%), moderate (25% to 49%), low (10% to 24%), and anything <10% would not be outside the 95%CI of zero.
- Page 1, lines 22-26: results regarding axial elongation are not mentioned in the abstract, but are later on reported in the manuscript.
Yes, we have tried to include the AL results in the Abstract but due to the strict 200 word limit imposed by JCM, we had to remove those results from the Abstract, as they refer to the secondary outcome of the trial.
Introduction:
- Page 1, line 44: These interventions are implemented not because of popularity, but due to their proven efficacy. Please rephrase.
We did not comment on the causes of the frequency of use of those treatments there but the fact that they are used most frequently in certain myopia clinics. We have substituted “popular” with “frequently used”.
- Page 2, lines 48-49: What were the drop-out rates for RCTs examining Ortho-K and multifocal lenses? Please provide numerical values and corresponding references.
Both published RCTs for Ortho-K had high drop-out rates: Cho & Cheung (2012) had 23.5% and Charm & Cho (2013) had 46.2% at the end of the trial. The rates are also listed in Table 3 of the IMI Interventions Report (Wildsoet et al. 2019). As we have stated in lines 307 – 309, 6 out of the 9 RCTs of soft MFCL that met our selection criteria exceeded the 20% of drop-outs threshold at the final follow-up, and 5 of them recorded >40% loss to follow-up. The 5 with >40% drop-outs were the 4 interventions of Sankaridurg et al. (2019) (recruited a total of 508, dropped out before 24M visit 274, i.e., 53.9%) and Lam et al. (2014)(recruited 221, of which 93 dropped out before the 24M visit, i.e., 42.1% drop-out rate), while the 6th trial that exceeded 20% threshold of drop-outs was Chamberlain et al. (2019)(144 randomised, 108 completed 36M visit and analysed, dropped out 36, i.e., 25%). We did not include these figures in the manuscript, as we do not feel they are relevant to our study – we have not run any meta-analyses of these interventions because there are not enough RCTs to warrant such meta-analysis.
- Page 2, lines 63-65: “To compare..+2.50D)” unclear phrasing.
We are unsure what is unclear in this sentence. In selecting the soft MFCL trials, we have excluded one trial which did not use a constant intervention (addition power of the lens) in the test group (Aller et al. 2016), as all the PAL trials used the same (“constant”) addition power for all patients in the treatment group. The range of addition powers of soft MFCL was twice as large than those used in the PAL trials with a range of 1.0 D vs. 0.50 D for PALs, which may have added to higher heterogeneity of this trial set.
- Page 2, lines 69-70: Authors acknowledge that RCTs for PALs have already been analyzed in previous studies. The following study should also be referenced: Prousali E, Haidich A, Fontalis A, et al (2019) Efficacy and safety of interventions to control myopia progression in children : an overview of systematic reviews and meta-analyses. BMC Ophthalmol 19:106
We were aware of this paper, but we admit that we have not originally examined the list of original clinical studies found by the search of databases by Prousali et al. (2019), which have been published in the on-line material attached to this publication. We have now reviewed the 44 original studies found in the database search carried out by Prousali et al. and listed in Appendix 3 of the referenced paper. No new myopia control trials of PALs that we were not aware of have been found, but we have added a reference to this paper in line 70, as suggested.
- Page 2, lines 74-76: Authors appear to comment on their findings in the Introduction section. Please include all findings under ‘Results’.
Well, this discussion relates to our initial attempt to understand the reasons behind relatively high heterogeneity levels reported by some of the already published meta-analyses of myopia control trials using PALs and other interventions. One of the possibilities considered was the choice of the outcome variable. All of the meta-analyses we have examined adopted the difference between the progression of SER or AL between the test group and the control group. However, it is well known that progression rates between cohorts of children of different ethnicities, urban/rural locations, ages, sometimes gender, parental myopia state and baseline refraction can vary. So, it is reasonable to expect that the difference between the progression in the test group and control group can also vary with a number of those factors. In order to compensate for such variability, meta-analyses are often carried out on scaled output variables. To make such a methodological decision before running any meta-analyses, we have just compiled a simple table of 4 high-quality trials and their reported outcomes using two different variables: absolute change in progression and a simple progression of the test group scaled by the progression of the control group. This table is shown below.
Trial |
Mean Age (Years) |
PAL Group Progression |
SVL Group Progression |
Change in Progression |
Scaled Progression |
COMET2 (2011) |
10.1 |
-0.29 |
-0.42 |
0.13 |
0.69 |
Berntsen (2012) |
9.9 |
-0.35 |
-0.52 |
0.18 |
0.67 |
Gwiazda (2003) |
9.3 |
-0.42 |
-0.60 |
0.18 |
0.70 |
Hasebe (2014) |
10.3 |
-0.56 |
-0.80 |
0.24 |
0.70 |
It can be seen that the control group progression rates and the actual changes in progression between the treatment and control groups in diopters in this set of trials vary almost by a factor of 2, but the progression in the treatment group scaled by the mean progression of the control group varies by less than 5% of the mean value. So, we were able to make a methodological decision on the variable to be analysed without doing any meta-analysis. Hence, we do not believe that the referenced sentence on p. 2 belongs to the Results section, as it needs to be used in the Methods section, which comes before the Results section. So, we have amended the Methods by inserting a brief description of this argument along with the table above after the Scaled Progression variable y is introduced (after line 172 on p.4 in the original manuscript). The new table has been named Table 1 and the remaining tables have been renumbered. We have also removed the reference to this pre-analysis from the Introduction.
We would like to add a note here that we have changed name of the variable y from “Scaled Progression Difference (SPD)” to “Scaled Progression (SP)”. We have realised that the word “Difference” is not needed here, as no difference is being calculated to get y. We have initially written the variable y in a different form to make its relationship to the scaled progression difference in percentages more obvious and later changed it to a simpler formula but forgot to adjust the wording. This has been now rectified throughout the text of the amended manuscript.
- Page 3, lines 108-116: Authors mention that they aim to analyze RCTs on PALs, which have already been included in previous published meta-analyses. The authors need to clearly explain how their analysis adds information to those already presented and convince the reader that this meta-analysis will provide benefit over what is written in the literature, ie. the recently published meta-analysis by Kaphle et al. (Kaphle D, Atchison DA, Schmid KL (2020) Multifocal spectacles in childhood myopia: Are treatment effects maintained? A systematic review and meta-analysis. Surv Ophthalmol 65:239–49.). Of note, in this paragraph there is no reference to multifocal soft contact lenses, which are supposed to be included in the analysis.
In the referenced lines we have provided the arguments of how our meta-analyses will provide benefit over those published in Huang et al. (2016), which has the largest overlap of PAL data used of all other published meta-analyses (7 out of 10 trials used in our meta-analysis – 70%). The advantages of our meta-analysis over Kaphle et al. (2020) can be easily deduced from the listing of its drawbacks on p.2, line 95 to p.3, line 107. Firstly, they have included multiple modalities of intervention in the main body of their meta-analysis, not just PALs. Secondly, they have included only 5 trials of PALs (50% of those we have analysed). In the 6M over 6M meta-analyses they have actually used only 3 PAL trials (Hasebe 2008, Berntsen 2012 and Edwards 2002). Thirdly, they did not use the outcome variable that provides lower heterogeneity. Fourthly, the 12M outcomes of the 6M over 6M analyses provide the mean absolute change in SER progression that is less than half of the value calculated at 12M from the 12M over 12M period meta-analysis, and they did not appear to make any attempt to explain a 100% inconsistency between the two meta-analyses they have presented at the same 12M follow-up. We have also added a sentence at the beginning of the Discussion section pointing out a different outcome of our meta-analysis of PAL trials in the first 12M follow-up compared to that of Kaphle et al. (2020). We have found no weakening of the effect between the first 6M and in the 6M to 12M interval, while Kaphle et al. (2020) analysis found the effect size more than halved in the second period.
The meta-analysis of soft multifocal lenses has only been done for comparison purposes because our meta-analysis could not be meaningfully compared to any other published meta-analysis due to the use of different variables.
Methods:
- Page 3, line 117: change to ‘Methods’.
Changed, as requested.
- Page 4, line 188: change to ‘Eligibity Criteria’.
Changed, as suggested.
- Page 5, line 204: remove ‘somewhat’.
Removed, as requested.
- Page 5, line 213: remove ‘have’.
Removed, as suggested.
- Page 5, lines 219-220: the authors should explain why vitreous chamber depth has been used as an equivalent of axial length and provide a relevant reference.
The vitreous chamber depth was used as a substitute of AL in one of the PAL trials included in the analysis because the authors of the paper chose to measure VCD instead of AL, and AL measurements were not available for that trial (Yang et al. 2009).
Justification of the equivalence can be found in our Reference [53], beginning of the last paragraph on p. 1342: “The vitreous chamber grew in a fashion similar to AL (Fig. 5).”
- Page 4-6, lines 199-264: this part should be added as a separate sub-section named ‘study selection’.
A separate sub-section “Study Selection” has been added.
- Page 6, lines 266-276: quality assessment should be added as a separate sub-section.
A new sub-section “Trial Quality Assessment” added.
- The authors provide no information regarding the search strategy, if implemented, for this review. Under the ‘Introduction’ section (Page 2, lines 66-70), authors mention that no systematic search was performed. This would be rather inappropriate for a high-quality study; authors should clearly mention in the text on which literature search strategy they have based their findings.
As was explained on page 2, lines 66 – 70, no new systematic searches were performed in this area, as the recent IMI Interventions Report and a range of referenced systematic reviews and meta-analyses recently published (two of them in 2020) have covered the area adequately, and we are convinced that no new studies of PALs for myopia control would be found, if such a search were performed again. This is underscored by the latest published over 220-page systematic review and meta-analysis of RCTs for myopia control published by Walline et al. (2020), which included only 5 trials of PALs and 4 trials of soft MFCLs in their meta-analyses. Our view is that the quality of meta-analysis is determined by the number of good quality studies included and consistency of their outcomes.
- The whole ‘Methods’ section needs shortening and statistical analysis should be preferably moved after ‘quality assessment’ sub-section. A separate, clear sub-section on outcome measures should also be added.
The first part was already addressed above. We are not sure what is meant by the movement of statistical analysis after quality assessment, as the statistical analysis is presented in the Results section.
We have added the assessment of outcome measures to the Method section, as discussed above.
Discussion:
- Page 15, line 474: do the authors wish to comment on a proposed mechanism for how part of axial elongation might not contribute to myopic progression?
We didn’t say that part of axial elongation does not contribute to myopic progression in the referenced line. We said that part of axial elongation is compensated by the crystalline lens thinning, as the latter reduces the focusing power of the eye and moves the focal plane of the eye backwards, and thus compensates at least part of the axial elongation. In juvenile emmetropes and stable myopes this process effectively cancels myopia progression. It must also be noted that we are talking here about the efficacy in terms of relative, not absolute values, i.e., percentage change in progression. We have added some additional comments to clarify this point after the line 477 of the original manuscript.
- Page 16, lines 509-510: remove ‘around’ and provide the exact mean values.
The provided values are not exact because a couple of trials did not report the mean age of their participants (Shih et al. 2001 and Hao et al. 2006). In those cases, we have approximated the mean age with the mid-point of the age range. Hao (2006) did not report the mean refraction at baseline either. In this case it was excluded from the calculation of the mean baseline refraction.
- Page 16, lines 518-544 and figures 7-8: results should be reported under the ‘Results’ section, rather than ‘Discussion’.
Lines 518 – 545 and Figures 7-8 have been moved to the Results section, new subsection 4.6.
- Page 18, lines 573-577: what do authors mean by ‘variable’?
We mean that the power of the rigid Ortho-K contact lenses used by the subjects in the test group was not constant through the duration of this 5-year trial (Hiraoka et al.2012) which led to the increase in asphericity of the cornea in this group, as the trial progressed. This is explained in the following lines: 577 – 582.
- ‘Discussion’ seems to lack a clear appraisal of the authors’ findings and discussion of their clinical significance, specifically with regard to establishing guidelines for myopic progression control. In general, ‘Discussion’ appears wordy and confusing, which would not help the readers absorb any meaningful conclusion.
We thank the Reviewer 1 for directing us to focus more on the impact of our research findings on the clinical guidelines for the control of myopia progression. Since none of the authors is an optometrist or an ophthalmologist, our original focus was on assessing the clinical evidence scientifically, rather than providing a clinical guidance to eye care practitioners. This has stimulated us to make a greater effort in covering this aspect in our manuscript. Some additional comments have been made in the Discussion section, and we have added a new Conclusions section at the end, which provides the summary of our findings and recommendations for the clinical guidelines arising from our findings.

Reviewer 2 Report
This is an interesting study and paper. Since the advantages and limitations of Bayesian meta-analysis might be not familiar to many readers they should be clearly presented in the methodology section.
At the end of the Discussion section, it should be clearly highlighted what new comes from this analysis, as the Conclusions section.
Author Response
Dear Reviewer 2,
Please find attached our responses to your comments on our manuscript attached.
Your sincerely,
Saulius Varnas
--------------------------
Reviewer 2
Comments and Suggestions for Authors
This is an interesting study and paper. Since the advantages and limitations of Bayesian meta-analysis might be not familiar to many readers they should be clearly presented in the methodology section.
We have added the requested description in the Methods section after the L156 of the originally submitted manuscript.
At the end of the Discussion section, it should be clearly highlighted what new comes from this analysis, as the Conclusions section.
The Conclusions section has been added at the end.
Reviewer 3 Report
Dear Authors,
I would like to point that Figure 7 and 8 and their analysis should be part of the results not part of the discussion. The results which are coming from the Figures 7 and 8 should be discuss in the part of discussion. In my opinion at the end of the work as the separate part conclusions should be present.
Author Response
Dear Reviewer 3,
Please find attached our responses to your comments on our manuscript attached.
Your sincerely,
Saulius Varnas
--------------------------------
Reviewer 3
Comments and Suggestions for Authors
Dear Authors,
I would like to point that Figure 7 and 8 and their analysis should be part of the results not part of the discussion. The results which are coming from the Figures 7 and 8 should be discuss in the part of discussion.
We have now described the meta-analysis of soft MFCLs in the new subsection 4.6 of the Results section and moved Figures 7-8 to this section. The comparative analysis of the PAL and MFCL efficacy performances has been expanded in the Discussion section and the new Conclusions section.
In my opinion at the end of the work as the separate part conclusions should be present.
The Conclusions section has been added at the end.